

# Quantifying Hail Size Distributions from the Sky: Application of Drone Aerial Photogrammetry

Joshua S. Soderholm[1], Matthew R. Kumjian[2], Nicholas McCarthy[3], Paula Maldonado[4], and Minzheng Wang[5]

[1]University of Bonn, Meteorological Institute, Germany
[2]The Pennsylvania State University, Department of Meteorology and Atmospheric Science, United States
[3]The University of Queensland, Australia
[4]University of Buenos Aires, Argentina
[5]Northraine PTY. LTD.

**Correspondence:** Joshua Soderholm (joshua.soderholm@bom.gov.au)

**Abstract.** A new technique, named "HailPixel," is introduced for measuring the maximum dimension and intermediate dimension of hailstones from aerial imagery. The photogrammetry procedure applies a convolutional neural network for robust detection of hailstones against complex backgrounds and an edge detection method for measuring the shape of identified hailstones. This semi-automated technique is capable of measuring many thousands of hailstones within a single survey, which is several orders of magnitude larger (e.g., $10,000$ or more hailstones) than population sizes from existing sensors (e.g., a hail pad). Comparison with a co-located hail pad for an Argentinan hailstorm event during the RELAMPAGO project demonstrates the larger population size of the HailPixel survey significantly improves the shape and tails of the observed hail size distribution. When hailfall is sparse, such as during large and giant hail events, the large survey area of this technique is especially advantageous for resolving the hail size distribution.

## 1 Introduction

Measurements of the hail size distribution (HSD) are challenging to collect owing to the infrequent and hostile nature of hailstorms. Because of these constraints, HSD measurements are uncommon, especially for larger hail ($> 25$ mm). Such observations are necessary to constrain hail microphysics parameterization schemes used in weather and climate models, and for hail detection and sizing algorithms from weather radar. Ground sensors for measuring the size distribution of large hail can be separated into those that provide time-recording (e.g., hail disdrometer) and those that provide time-integrated measurements (e.g., hail pad). Time-recording instruments such as impact or optical disdrometers provide valuable information on the temporal variability of the HSD within a given storm, but are often expensive to fabricate and maintain, and difficult to deploy. Thus, such instruments typically are only deployed as smaller networks or for field campaigns (e.g., Federer and Waldvogel,





1975; Brown et al., 2014). In contrast, time-integrated instruments often are cheaper to fabricate, maintain, and deploy, making them attractive options for longer-term monitoring of hailfall. The most commonly used time-integrated instrument for measuring HSDs is a hail pad, consisting of a foil covered styrofoam pad that preserves dents of hail impact (Long et al., 1979). This sensor is cost effective and has seen extensive use by previous and ongoing campaigns in the US and Europe over the

last 50 years (Cheng and English, 1983; Fraile et al., 1992; Cifelli et al., 2005; Kalina et al., 2014). Both hail pads and hail disdrometers provide reasonable estimates of hail size, but are subject to significant limitations even with careful calibration (e.g., Palencia et al., 2011). Further, both time-recording and time-integrated instruments for measuring the HSD utilize a small sample area on the order of 0.1 to 0.3 $m^2$. Towery et al. (1976) suggests this small sample area is likely to underrepresent the HSD, particularly for larger hail, and recommends deployment of multiple sensors to minimize this effect.

The concentration of large hail, and particularly giant hail ($> 100$ mm) can be very sparse (Witt et al., 2018), severely limiting the effectiveness of these small ground sensors even with multiple units (Browning, 1977). To overcome these sampling limitations, we describe a new time-integrated technique for measuring the HSD by combining aerial imagery captured from a small unmanned aircraft with deep learning and computer vision feature extraction. Methods involving Machine Learning and Artificial Intelligence have seen increased utilization in the atmospheric sciences community, including for the application of

severe weather (e.g., McGovern et al., 2017; Gagne et al., 2019); however, there has been limited usage of such methods in targeted field observation datasets and in situ data. The new technique described here, named "HailPixel," enables the capture of very large areas ($> 1500$ $m^2$, equivalent area to several thousand hail pads) immediately following a hailstorm. This paper describes the methods of imagery capture and semi-automated extraction of the HSD using a combination of deep learning and computer vision techniques. Results from a HailPixel survey of a hailstorm on 26 November 2018 in San Rafael (Argentina)

are discussed in the context of existing studies and potential improvements for future surveys.

## 2   Data and Approach

Aerial imagery surveys of hail coverage were conducted in the Mendoza Provence of Argentina after hailstorms on 25 and 26 November 2018 during the RELAMPAGO field campaign (Nesbitt, 2019). To effectively extract the HSD from aerial imagery, hailstone size must of sufficiently larger than the effective ground resolution of the sensor and the concentration of hailstones

must be sufficiently low so that overlapping stones are minimised. These requirements were satisfied only for the 26 November event, and imagery from this event will be used throughout the paper. The hail swath observed from the 26 November event was produced by a marginally supercellular storm that developed in an environment of moderate instability and deep-layer shear. The storm initiated on the Andean Mountains and tracked approximately 120 km east-northeast towards the city of San Rafael before observations were made. A single hail pad ($300 \times 400 \times 30$ mm polystyrene foam block covered in aluminium

foil) was also deployed 2 km southwest of the aerial survey site for the San Rafael hailstorm, providing a secondary measure of the HSD. To estimate hail size from hail pad indentations, the major and minor axis length of individual dents was measured with digital calipers and transformed into hail major and minor axis size using a relationship developed by the Community Collaborative Rain, Hail, and Snow Network (N. Doesken, personal communication, April 17, 2019).





## 2.1 Imagery

A DJI Phantom 4 Pro V2 aircraft and Pix4DCapture flight-control software was used for image acquisition. The integrated aircraft camera uses a $13.2 \times 8.8$ mm CMOS sensor which provides 20M effective pixels, and an auto-focus lens with a focal length of $8.8 - 24$ mm and maximum field of view of $84°$. For the 26 November event, the aircraft was flown at an altitude of 10 m (relative to the take-off location) over a rectangular survey area of 1290 m$^2$, providing a $\sim$2.7-mm ground sampling distance ((Fig. 1a). Images were captured with a $70\%$ overlap laterally and medially at a flight speed of 1 m s$^{-1}$ over a surface consisting of sparse grasses, small shrubs, gravel, and dirt. A large image overlap and slow flight speed was selected to increasing the number of quality matching points during orthomosaic construction and reduce motion blur (Bemis et al., 2014). The survey was initialized immediately once hail fall concluded and required approximately 4 minutes to complete. The location of images was measured using the integrated GPS receiver, which has an accuracy of $\pm 1.5$ m. Precise location measurements (e.g., real-time kinematic positioning) are not essential for improving the pixel size accuracy during photogrammetry processing (Strecha, 2012).

The Pix4DMapping software package was used to generate orthomosaic imagery and a digital elevation model (DEM) from the survey photos (Strecha, 2012). The software is based on the Structure from Motion photogrammetry technique and uses the following automated steps:

1. Tie points between the survey images are identified. Each tie point must be matched in at least 3 images

2. Tie points are combined with positioning and orientation information from the aircraft autopilot to reconstruct the camera perspective and position for each survey image. This information is used to verify the quality of matching points and calculate the 3D coordinates of tie points.

3. The sparse point cloud of 3D coordinates is interpolated to obtain a gridded DEM.

4. The DEM is used to project every image pixel and to calculate a orthomosaic.

An average of $181,081$ matched tie points were found per m$^3$ with a mean geolocation error of less than 1 mm. Analysis of the DEM indicates a gradual slope was present across the survey area with a total change in elevation of approximately 2.3 m (not shown). Two scale markers consisting of $300 \times 300$ mm black and white vinyl tiles were also placed into the aerial survey area to provide a secondary check of pixel size (2.5 mm) within the orthomosaic.

## 2.2 Hail Detection

To efficiently identify the many thousands of hailstones captured in the aerial imagery after the San Rafael hailstorm, automated feature detection techniques were explored. Simple thresholding of pixel luminosity for detecting hailstones performed poorly owing to similar luminosity from sparse grasses, light dirt patches, pale colored rocks and leaf debris, and for instances where hailstones were in contact. Despite the low contrast, hailstones were easily identifiable in the imagery by human observers, motivating the application of a convolutional neural network (CNN) model. This class of machine learning algorithms develops complex feature recognition filters independent of prior knowledge, inspired by processes within the animal visual cortex



(Hubel and Wiesel, 1968). Over the last two decades, CNN's have become a rapidly developing research tool that excels at image recognition (e.g., Razavian et al., 2014; Krizhevsky et al., 2012). For hail detection, the state-of-art Mask Region-CNN (R-CNN) model was implemented (He et al., 2017). This technique combines the optimised selection and parallel processing of proposed feature regions (Fast R-CNN) with semantic segmentation, whereby each pixel is classified. Mask R-CCN

architecture and implementation used is described in detail by He et al. (2017).

To reduce memory requirements, the 489-megapixel aerial survey orthomosaic was divided into 1961 tiles of size 600 $\times$ 600 pixels, including a 50-pixel overlap along edges with neighbour tiles to avoid cropped hailstones (Fig. 1b, 2a,b). For training the Mask R-CNN model, 12 tiles were selected to represent the varying background conditions and were manually annotated using the VGG Image Annotator (VIA) tool (Dutta and Zisserman, 2019). Nine tiles were randomly selected for

training (containing 729 annotated hailstones) and the remaining three for validation (Fig. 11c). The Mask-RCNN training was initialized with the pre-trained weights from the Microsoft COCO dataset (set of $> 2 \times 10^5$ labelled images; Lin et al. (2014)), which capture many features in natural images. Utilizing these weights greatly reduces the training time required to recognize hail. The default learning and weighting configuration described by He et al. (2017) were applied and training was performed on 8 GPUs with 1 image per GPU for 3,000 iterations ($\sim$43 minutes of computation time). The trained model detected more

than 94% of hailstones in each validation tile with a false alarm rate of $< 1\%$. When applied to all tiles, the trained Mask R-CNN model detected a total of $46,871$ hailstones.

## 2.3   Hail Size Measurement

The segmentation mask generated by the Mask R-CNN model was initially tested for hail size measurement, but found to contain small errors that rendered it unsuitable. To provide the pixel-level accuracy required for measuring hailstones, an edge

detection algorithm was developed to find the steep "lightness" gradient[1] at the hailstone edge that occurs radially from the hailstone centroid (Figure 1d). The HSL color space is an alternative to Red-Green-Blue (developed for color displays) that is commonly used in computer vision applications for reducing the correlation between colors (Cheng et al., 2001). The hailstone centroids required to initialize the edge detection technique are derived from the segmentation mask. Two additional quality control steps are also applied to the centroids and image tiles:

1. Tiles where hail was obscured (e.g., under long grass or shrubs) or water had accumulated were removed, leaving 188 "clean" image tiles containing $15,983$ hailstones over a total area of $305.6$ m$^2$ for hail size measurement.

   2. Hail centroids for the 188 clean image tiles were manually assessed and amended if required using the VIA annotation tool.

The clean image tiles are next transformed into the HSL color space and the hailstone size is measured for every centroid using

the following procedure. First, coordinates of 12 equally spaced radials of length 20 pixels from the centroid ($p_0$) are calculated (Fig. 2c), denoted as $p_i^k$ Where $i$ is the pixel index ($i = 1,...20$) and $k$ is the radial index ($k = 1,...,12$). For all points along a

---

[1]lightness here is from the Hue-Saturation-Lightness (HSL) color space





radial, the lightness values $L(p_i^k)$ are extracted. Then, the gradient of lightness values $L'(p_i^k)$ along each radial are calculated. Starting from the centroid of each radial, the edge point is found at coordinate $p_i^k$ when the following criteria are met:

$$L'(p_i^k) < 0.75 \times L'(p_{i-1}^k)$$

and

$$L(p_0) - L(p_i^k) > 50$$

Once all edge points are found along the radials (Figure 2d), the median distance $\widetilde{d}$ from the centroid is calculated for each edge point. If an edge point falls outside the range $[\widetilde{d} \times 0.5$ to $\widetilde{d} \times 1.5]$, it is replaced by $\widetilde{d}$. Finally, to measure the major and minor axis length of the hailstone, the minimum bounding box (allowing for rotation) is calculated for the set of edge points.

## 3   Results and Discussion

The resulting distribution of major axis length and axis ratio for the San Rafael hailstorm is shown in Figure 4, along with the distributions obtained from the hail pad. Comparison of the major axis length distribution from the HailPixel and hail pad techniques clearly demonstrate the value of aerial photogrammetry: the large population size ($n = 15\,983$) of the aerial survey provides a defined distribution shape and tails (Figure 4). The HailPixel distribution peak is 2.5 mm lower than the hail pad peak, possibly due to melting of hail on the ground before it was photographed or uncertainty in hail size retrievals.

The distribution shape is well approximated by a Gamma probability distribution function (PDF) with a mostly absent lower quartile and long upper tail. The gamma PDF was also found to be most suited for major axis length in a number of other case studies studies, including Ziegler et al. (1983) for Okalahoma (US), Wong et al. (1988) for the Albert (Canada), and Fraile et al. (1992) for León (France).

The distribution of axis ratios from the HailPixel survey is well approximated by an exponentially increasing function (not

shown); however, a local maximum at 0.8-0.85 suggests a more complex underlying distribution. Note that oblate hailstones are most likely to rest on the surface with their minor axis orientated vertically; thus, the HailPixel technique would not measure the true minor axis length in this scenario, but rather provides estimates of the intermediate axis ratio (assuming ellipsoidal geometry for hailstones). This limitation is likely to also effect hail pad measurements when tumbling motions of oblate hailstones are not too extreme. Comparing the HailPixel intermediate/major axis distribution with Giammanco et al.

(2014) major/minor axis distributions demonstrates the expected skew towards higher axis ratios in the HailPixel dataset.

Both HailPixel and hail pad data demonstrate decreasing axis ratio with increasing hail size. This shape of this relationship becomes apparent when the highly variable HailPixel data are binned into 5-mm intervals. For the 20-30-mm hail size range, axis ratio remains constant and close to 0.9. For sizes > 30 mm, axis ratio decreases by $0.4-0.5\%$ mm$^{-1}$. Despite the potential bias in axis ratio measurements, the shape of this trend is comparable to observations by Knight (1986) for an Alberta (Canada)

hailstorm. Another study using a 3-year database of hailstones collected from the Great Plains by Giammanco et al. (2014) demonstrates a less significant decreasing trend between hail size and axis ratio, and less spherical stones for smaller sizes. It





is likely that this relationship is also highly variable between hailstorm cases and within hailstorms (Federer and Waldvogel, 1975; Ziegler et al., 1983; Knight, 1986).

It is also important to highlight the optimal conditions and configuration for future HailPixel surveys. We recommend avoiding inhomogeneous background surfaces if possible, with cut or grazed turf grasses being most ideal. A uniform and

contrasting background will likely permit the use of less complex hail detection and sizing techniques. Large survey areas ($> 1000$ m$^2$) are only necessary when very sparse giant ($> 100$ mm) hailstones are present. Assuming a normally distributed sample mean, a sample size of 2088 hailstones is required to represent the population mean (from $15,983$ hailstones) within a $2\%$ confidence level at the $95\%$ significance level. This sample size equates to a sample area of $40.1$ m$^2$ for the 26 November survey; however, an area of at least $\sim250$ m$^2$ is recommended to adequately resolve the tails of the distribution. Higher-

resolution imagery would allow for small ($< 20$ mm) hailstones to be measured, but the increasing susceptibility to motion blur would likely require the aircraft to remain stationary during image capture. To further quantify the measurement uncertainty from the HailPixel technique, we recommend that hail within a $1$ m$^2$ area of the aerial survey is manually measured for three orthogonal axes immediately following aerial capture. Finally, minimizing the melting of hailstones is critical. Where possible, avoid aerial surveys of areas where water may flow or accumulate and conduct surveys immediately after hail fall ceases.

**4   Summary**

This paper describes the novel HailPixel aerial photogrammetry technique for measuring time-integrated hail size distributions after cessation of hail fall. The workflow for collecting imagery, detecting hail stones, and measuring hail size is described, including the use of state-of-art mask R-CNN image segmentation algorithm. Results from a HailPixel survey after the 26 November 2018 San Rafael (Argentina) hailstorm are compared with observations from a co-located hail pad. Despite potential

bias of axis ratio measurements, the HSD distributions and relationships observed for the San Rafael hailstorm are comparable to previous studies, although with substantially larger population sizes. Further HailPixel surveys are encouraged to quantify the variability of these distributions, particularly for hailstorms producing sparse giant hail. Ongoing work to relate HailPixel results with mobile polarmetric radar observations from the 26 November 2018 San Rafael hailstorm will explore the signatures of hail size and swath extent for this event.

*Data availability.*   All imagery, hail pad and hail retrieval data used for this work are publicly available: Soderholm. (2019). HailPixel Survey Data and Analysis from 26 November 2018, San Rafael, Argentina [Data set]. Zenodo. http://doi.org/10.5281/zenodo.3383227

*Author contributions.*   Soderholm designed the HailPixel technique and Soderholm, Kumjian, McCarthy and Maldonado applied the technique following hailstorms during the RELAMPAGO project. Soderholm developed the hail identification and sizing algorthims with assistance from Wang. Soderholm prepared the manuscript with contributions from all co-authors.



*Competing interests.* The authors declare that they have no conflict of interest.

*Acknowledgements.* The authors wish to thank the Australian National Environmental Science Project (NESP) for funding the HailPixel project and the RELAMPAGO project for the unique opportunity to operate HailPixel surveys in Argentina. Initial testing, methodology reviews and equipment loans were provided by the School of Earth, Atmosphere and Environment, Monash University; in particular, Sam
5  Thiele and Steven Micklethwaite. The first author is supported by the Alexander von Humboldt Foundation and AWS Earth on AWS Cloud Credits for Research program. The second author is supported by NSF Grant AGS-1661679 and an award from the Insurance Institute for Business and Home Safety.



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

**Figure 1.** Workflow of (a) data collection, (b) tiling of orthomosaic, (c) hail detection using the Mask-RCNN technique and (d) hail size measurement using radial transects.

**Figure 2.** Demonstration of hail size extraction from the 26 November 2018 survey orthomosaic (a) from a single tile (b; blue bounding box) for a single hail stone (c, black circle in b). Radial transects for extracting imagery lightness are shown in (c) as black lines radiating from hailstone centroid (blue marker) and hailstone edge pixels along transects are numbered. Subplot (d) shows the normalised pixel lightness along the 12 transects shown in (c) with the corresponding edge pixels marked.



**Figure 3.** Orthomosaic RGB imagery from the 26 November 2018 survey overlaid with the outlines of tiles used for the extraction of hail size distribution statistics



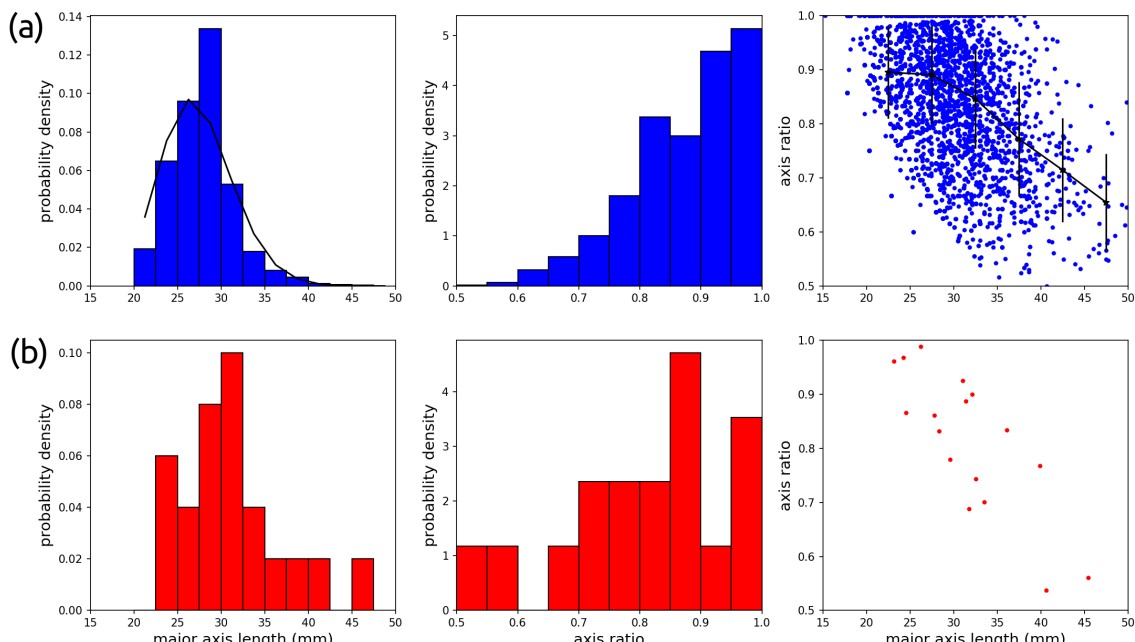

**Figure 4.** Distribution of hail major axis length, minor axis ratio and scatter plot of axis ratio and major axis length from the (a) photogrammtery and (b) pad hail size retrievals for the 26 November 2018 survey. A fitted Gamma distribution probability density function for photogrammtery major axis size distribution is shown (black line). Photogrammtery scatter plot observations are binned using 5-mm bin sizes and error bars represent ±1 standard deviation.