# Peer review of "Quantifying Hail Size Distributions from the Sky: Application of Drone Aerial Photogrammetry"

_Atmospheric Measurement Techniques, 2019_

## Referee Comment (RC1) · Anonymous Referee #2 · 19 Sep 2019

The authors present a new technique for measuring hail size distribution using aerial photography and a machine learning approach to detecting and measuring individual hailstones. This is a novel method using new technologies and would provide better sampling with more automation.

There are, however, a number of requirements for this technique and I would like to see more discussion on these limitations and in what kind of scenarios this method would be practical.

Further concerns:

Pg 2 Ln 22-25: Can you quantify these requirements better? It looks like, in this case, the hailstones must be larger than 20mm. What kind of spacing must there be between

them on the ground? How much is the technique affected by the presence of smaller hailstones as well?

Pg 3 Ln 6: Is there a restriction on the 10m wind speed in order to fly the drone at such a low speed? Does accuracy fall off with a large or variable wind?

Pg 4 Ln 8: How were the 12 training tiles chosen? Does the number of tiles necessary for training depend on the concentration of hail, or on the type of background?

Pg 4 Ln 26: Do the tiles have similar distributions or concentrations?

Pg 5 Ln 5: What kind of range does the 'lightness value' have? If the lightness value of a hailstone must be >50 more than that of the edge, how much does this restrict the type of background against which hail can be measured? How sensitive is the lightness value or its variability to the overhead light (sky conditions, sun angle)?

Pg 5 Ln 10: How many hailstones were counted on the pad? How does the total concentration compare? Were there no hailstones <20 mm measured on the hailpad or were they just not considered for this comparison?

---

## Author Comment (AC1) · 29 Oct 2019

Interactive comment on "Quantifying Hail Size Distributions from the Sky: Application of Drone Aerial Photogrammetry" by J. S. Soderholm et al.

Anonymous Referee #2

The authors present a new technique for measuring hail size distribution using aerial photography and a machine learning approach to detecting and measuring individual hailstones. This is a novel method using new technologies and would provide better sampling with more automation. There are, however, a number of requirements for this technique and I would like to see more discussion on these limitations and in what kind of scenarios this method would be practical.

[Figure]

The authors wish to thank the reviewer for taking the time to read this paper and the valuable comments regarding improvements to the discussion of requirements and limitations.

Further concerns:

Pg 2 Ln 22-25: Can you quantify these requirements better? It looks like, in this case,the hailstones must be larger than 20mm. What kind of spacing must there be between them on the ground? How much is the technique affected by the presence of smaller hailstones as well?

####################

This is an important point, thank for for highlighting it. The hail size requirements are a dependent on the effective ground resolution of the orthomoasic, and thus are best discussed after the sensor is introduced (section 2). Two additional sentences have been added to section 4 (p. 6 lines 17-20) that provide recommended minimum hail size and ground coverage for successful HailPixel surveys. In summary this is a minimum value of 20 mm for the major axis length and a maximum of 50 % ground coverage. Further testing would be required to determine the effect of smaller hailstones on the measurement accuracy.

Pg 3 Ln 6: Is there a restriction on the 10m wind speed in order to fly the drone at such a low speed? Does accuracy fall off with a large or variable wind?

####################

Thank you for raising this point. The wind speed at the time of capture was a gentle breeze (3.5 – 5.5 m/s) and this has been added to page 3, lines 6-7. The authors recommend that hailpixel surveys are conducted with near surface below 8 m/s to reduce the likelihood of motion blurring. This recommendation has been added to the text in page 6, lines 22-23. The authors believe the accuracy of the hail size retrieval would be effected without a doubt in stronger winds.

[Figure]

Pg 4 Ln 8: How were the 12 training tiles chosen? Does the number of tiles necessary for training depend on the concentration of hail, or on the type of background?

####################

Twelve tiles were chosen to ensure the total number of stones was approximately 1000. A smaller sample size may have also been appropriate but the sensitivity of the model performance to training size was not explored. Further, pre-training weights from the COCO dataset were used to initialise the model, limiting the need for very large training datasets. The specific tiles used for model training were manually selected to represent the different background types, as this is also an important part of training an RCNN model. The manuscript has been amended to clarify that the 12 tiles used used to achieve a sufficient sample size and were selected manually to sample the different background types (page 4, lines 11-12).

Pg 4 Ln 26: Do the tiles have similar distributions or concentrations?

####################

The hail concentration of individual tiles varied between 15 and 91 stones per m2, with a mean of 47 stones per m2. Inspection of the spatial distribution indicated that vegetation density and slope plays an important role in the concentration of hail when it finally comes to rest. Higher concentrations appear in regions of denser grass, likely due to the grassy areas acting to dampen bouncing and rolling of stones, increasing collection. The lowest concentrations appear on unvegetated areas where hail can readily bounce and roll. An analysis of individual tile concentration has not been included in the text because it doesn't reflect the true distribution of hailstones for these reasons.

Pg 5 Ln 5: What kind of range does the 'lightness value' have? If the lightness value of a hailstone must be >50 more than that of the edge, how much does this restrict the type of background against which hail can be measured? How sensitive is the lightness value or its variability to the overhead light (sky conditions, sun angle)?

####################

The lightness value is a 8 bit index that has a range of 256 values (added to the text in the foot note on page 4). The authors agree that the minimum difference in lightness between the hailstone centre and hailstone edge is definite limitation for the technique. However, even for light-coloured soils, the lightness difference still remained well above 50 units. A comments has been added to the text (page 5, lines 9-10) regarding the performance of this lightness difference threshold for different background types. Regarding the sensitivity of lightness values to the sky conditions, the November 2018 survey was conducted during cloudy conditions. The automatic exposure and white balance control on the UAV camera was able to compensate for the low lighting conditions. Comparable lightness values for hail and background types would be expected for full-sun conditions with the correct exposure and white balance adjustments.

Pg 5 Ln 10: How many hailstones were counted on the pad? How does the total concentration compare? Were there no hailstones <20 mm measured on the hail pad or were they just not considered for this comparison?

####################

Thank you for raising these important points. A total of 17 impacts with major and minor axis measurements were sampled from the hail pad, with a concentration of 141 stones per m2. This information has been added to the text on page 5 lines 15-16 and lines 23-25 respectively. The mean concentration observed by the HailPixel survey was 47 stones per m2, significantly less than the hail pad. This is possibly due to the fact the two samples were not co-located, and the hail pad experienced a longer duration of hail fall. Further, bouncing hail stones may have introduced secondary impacts on the hail pad that were indistinguishable. This justification has been added on page 5 lines 23-25. No stones less than 24 mm major axis length were sampled by the hail pad.

Please also note the supplement to this comment:
https://www.atmos-meas-tech-discuss.net/amt-2019-281/amt-2019-281-AC1-supplement.pdf

**Supplement:**

[revised manuscript text omitted]

---

## Referee Comment (RC2) · Tomeu Rigo (Referee) · 11 Dec 2019

Dear authors,

your manuscript presents a novelty in hail size research, taking profit of capabilities of drones and photography treatment techniques. Having in mind the review criteria of the journal (https://www.atmospheric-measurement-techniques.net/peer_review/review_criteria.html), I'm considering that it has an excellent scientific significance and good scientific and presentation qualities.

Besides, the paper answers positively to all the questions made in the same web.

However, I think that there are some points that you must solve before the acceptance of the paper.

[Figure]

- There are some typos: L1: "HailPixel." (the dot must be placed after the ") L6 of the page 3: ((Fig. 1a) - remove one (

- I think that your technique can be useful for more aspects that the cited in the text: for instance, for identifying the whole area affected by hail. I understand that your technique can discriminate between hail/non-hail pixels and then, you can delimitate the hailpath. In the same way, do you think that this is applicable in real-time? If your anwer is positive, explain it in the text, because this could help in many fields, in those areas commonly affected by hail events, such identification of damaged agriculture production or for insurance interests, among others.

- When you introduce hail-pads, you forget to mention automatic hail-pads (see, e.g. Martin Löffler-Mang, Dominik Schön, Markus Landry, Characteristics of a new automatic hail recorder, Atmospheric Research, V. 100, Issue 4, 2011, Pp. 439-446, ISSN 0169-8095, https://doi.org/10.1016/j.atmosres.2010.10.026.)

- In data and approach, please provide numbers (L25 pg 2): which size and densitity can be considered as thresholds?

- Where is the hail-pad used for the comparison located? You should indicate in a figure

- It results difficult to me understand which is the final size of the pixel, the one you use in fig 2c

- Those parts of the manuscript that are not referring to your work should be moved to the introduction, where the state-of-art is presented: e.g. L 1-5 of page 4, or some previous results used in your discussion.

- León is not placed in France (L18 Page 5)

And some final considerations:

- I think that you could do an effort and give more applicabilities to your research, such

the cited previously in my repport, including some references about this point (Botzen, W. J. W., Bouwer, L. M., & Van den Bergh, J. C. J. M. (2010). Climate change and hailstorm damage: Empirical evidence and implications for agriculture and insurance. Resource and Energy Economics, 32(3), 341-362. // Changnon, S. A., Changnon, D., Fosse, E. R., Hoganson, D. C., Roth Sr, R. J., & Totsch, J. M. (1997). Effects of recent weather extremes on the insurance industry: major implications for the atmospheric sciences. Bulletin of the American Meteorological Society, 78(3), 425-436. // Sánchez, J. L., Fraile, R., De La Madrid, J. L., De La Fuente, M. T., Rodríguez, P., & Castro, A. (1996). Crop damage: The hail size factor. Journal of Applied Meteorology, 35(9), 1535-1541. // Hohl, R., Schiesser, H. H., & Aller, D. (2002). Hailfall: the relationship between radar-derived hail kinetic energy and hail damage to buildings. Atmospheric Research, 63(3-4), 177-207.)

- In my opinion, you need to separate more clearly the part of your work from other previous techniques, and, besides, to present, maybe in a table, the technical characteristics of the analyzed imagery

- In the summary, you have to emphasize the advantages (also the disadvantages) of your technique in front of the current ones. In fact, you are saying that the hail distribution presented in your study is comparable with others, but in my opinion is clearly more accurated. You have to give weight to this point.

Best regards

---

## Author Comment (AC2) · 6 Jan 2020

Referee: Tomeu Rigo

Dear authors, your manuscript presents a novelty in hail size research, taking profit of capabilities of drones and photography treatment techniques. Having in mind the review criteria of the journal, I'm considering that it has an excellent scientific significance and good scientific and presentation qualities. Besides, the paper answers positively to all the questions made in the same web. However, I think that there are some points that you must solve before the acceptance of the paper

Hello Dr Tomeu Rigo, Thank you for the kind words and constructive review! Your feedback is much appreciated.

[Figure]

Question 1: There are some typos: L1: "HailPixel." (the dot must be placed after the ")
L6 of the page 3: ((Fig. 1a) - remove one (

Reply: After some investigation we have confirmed that the comma is to be placed inside the quotes on L1, page 1 (https://www.grammarly.com/blog/quotation-marks/). The additional bracket from L6, page 3 has been removed.

Question 2: I think that your technique can be useful for more aspects that the cited in the text:for instance, for identifying the whole area affected by hail. I understand that your technique can discriminate between hail/non-hail pixels and then, you can delimitate the hail path. In the same way, do you think that this is applicable in real-time? If your answer is positive, explain it in the text, because this could help in many fields, in those areas commonly affected by hail events, such identification of damaged agriculture production or for insurance interests, among others.

Reply: This is a very interesting idea to sample the hail swath extent/coverage rather than the hail size distribution. To sample at the resolution required for retrieving hail size, we had to fly the drone very low to the ground (10m), limiting the sample area to a few hundred meters. To collect aerial imagery that covers a significant portion of a hail swath (or say a farm or suburb), you would probably need to use a fixed wing drone, which provides greater endurance, flight speed and altitude performance than a quadcopter. Real-time extraction of information from the imagery would be more challenging, but not impossible. The authors thank the reviewer for this suggestion but believe this is outside the scope of this paper and expect it will be attempted at some point by hail researchers.

Question 3: When you introduce hail-pads, you forget to mention automatic hail-pads (see, e.g.Martin Löffler-Mang, Dominik Schön, Markus Landry, Characteristics of a new automatic hail recorder, Atmospheric Research, V. 100, Issue 4, 2011, Pp. 439-446, ISSN0169-8095, https://doi.org/10.1016/j.atmosres.2010.10.026.)

Reply: The reference to Löffler-Mang 2011 description of an impact disdrometer design

has been added to L3, p2

Question 4: - In data and approach, please provide numbers (L25 pg 2): which size and density can be considered as thresholds?

Reply: A lower threshold for the hail diameter is now suggested in L33 p2 and p6 L23. Providing an estimate for a maximum concentration threshold is not feasible as it depends on the hail size distribution. The author's suggestion of 30% coverage on p6 L24 should provide sufficient guidance for the concentration limits.

Question 5: - Where is the hail-pad used for the comparison located? You should indicate in a figure.

Reply: The authors believe showing the location of the hail pad and hail survey location on a map would have little value to the article, as the focus is not on the physical setting or spatial variability, but the technique. The exact location of the hail pad has been added in L5 p3, and the location of the aerial survey in L14 p3 for reference.

Question 6: - It results difficult to me understand which is the final size of the pixel, the one you use in fig 2c

Reply: The final pixel size of the orthomoasic has been added to the paper in L24 p3 to clarify the pixel size used in the analysis.

Question 7: - Those parts of the manuscript that are not referring to your work should be moved to the introduction, where the state-of-art is presented: e.g. L 1-5 of page 4, or some previous results used in your discussion.

Reply: Thank you for this suggestion. L1-5 of p4 have been moved to the introduction in p2 L16-20. For the discussion, the citations to other work are an essential part of this section and the author feels that duplicating or moving these citations in the introduction wouldn't add much value for the reader.

Question 8: - León is not placed in France (L18 Page 5)

[Figure]

Reply: Thank you, this has been corrected to Spain :)

Question 9: - I think that you could do an effort and give more applicabilities to your research, such the cited previously in my repport, including some references about this point Botzen,W. J. W., Bouwer, L. M., & Van den Bergh, J. C. J. M. (2010). Climate change and hailstorm damage: Empirical evidence and implications for agriculture and insurance. Resource and Energy Economics, 32(3), 341-362. Changnon, S. A., Changnon, D.,Fosse, E. R., Hoganson, D. C., Roth Sr, R. J., & Totsch, J. M. (1997). Effects of recent weather extremes on the insurance industry: major implications for the atmospheric sciences. Bulletin of the American Meteorological Society, 78(3), 425-436. Sánchez,J. L., Fraile, R., De La Madrid, J. L., De La Fuente, M. T., Rodríguez, P., & Castro,A. (1996). Crop damage: The hail size factor. Journal of Applied Meteorology, 35(9),1535-1541. Hohl, R., Schiesser, H. H., & Aller, D. (2002). Hailfall: the relationship between radar-derived hail kinetic energy and hail damage to buildings. Atmospheric Research, 63(3-4), 177-207.)

Reply: Thank you for this suggestion. An additional sentence has been added to the introduction to highlight the application of improving hail size distribution knowledge. Citations to Changnon et al. 1997, Sanchez et al. 1996 and Hohl et al. 2002 has been added to support this. P1 L15-17.

Question 10: - In my opinion, you need to separate more clearly the part of your work from other previous techniques, and, besides, to present, maybe in a table, the technical characteristics of the analysed imagery

Reply: Thank you for making this point. Two additional sentences have been added to the summary to highlight that this technique provides a significantly larger hailstone sample size that leads to more robust statistics (p7 L7-9). These sentences include the sample area and sample size of the analysed imagery.